# Virucidal and Antibacterial Chitosan–NanoCu Film-Coating-Based Technology: Complete Analysis of Its Performance on Various Surfaces

**DOI:** 10.3390/v17101347

**Published:** 2025-10-07

**Authors:** Victoria Belen Ayala-Peña, María Julia Martin, Jessica Otarola, Florencia Favatela, Jimena Soledad Gonzalez, Ana Lucía Conesa, Cybele Carina García, Claudia Soledad Sepúlveda, Vera Alejandra Alvarez, Verónica Leticia Lassalle

**Affiliations:** 1Departamento de Biología, Bioquímica y Farmacia, Universidad Nacional del Sur, Buenos Aires B8000, Argentina; vayala@criba.edu.ar (V.B.A.-P.); ana.conesa.97@gmail.com (A.L.C.); 2Consejo Nacional de Investigaciones Científicas y Técnicas (CONICET), Buenos Aires B1001, Argentina; julia.martin@uns.edu.ar (M.J.M.); jessica.otarola@uns.edu.ar (J.O.); florencia.favatela@outlook.es (F.F.); jimena.gonzalez@fi.mdp.edu.ar (J.S.G.); cygarcia@qb.fcen.uba.ar (C.C.G.); claudia@qb.fcen.uba.ar (C.S.S.); alvarezvera@fi.mdp.edu.ar (V.A.A.); 3Departamento de Química, INQUISUR, Universidad Nacional del Sur, Bahía Blanca B8000, Argentina; 4Facultad de Ingeniería, INTEMA, Universidad Nacional de Mar del Plata (UNMdP), Mar del Plata B7600, Argentina; 5Departamento de Química Biológica, Facultad de Ciencias Exactas y Naturales, Universidad de Buenos Aires, Buenos Aires B10001, Argentina

**Keywords:** antiviral, chitosan, surfaces, CuO nanoparticles

## Abstract

The transmission of viruses and bacteria via surfaces remains a persistent challenge for healthcare systems, leading to high public health costs and significant environmental impact due to the widespread use and disposal of single-use products. This study aims to evaluate the feasibility of using surface-covering films, based on biopolymers and inorganic nanoparticles, with strong antiviral and antibacterial properties, as a strategy to prevent infection transmission while offering a sustainable alternative to disposable materials. To this end, we developed a sprayable chitosan-based solution embedded with copper oxide nanoparticles (CH.CA@Cu). The solution demonstrated antibacterial activity against both Gram-positive and Gram-negative bacteria as well as virucidal activity, predominantly within one minute of exposure, against a wide range of viruses. After spraying various materials, the resulting film surfaces exhibited excellent adherence and uniform coverage, maintaining their integrity after contact. A field trial conducted in high-traffic environments confirmed the coating’s effectiveness. This long-lasting antiviral action supports their implementation, since the coated surface can continuously deactivate viruses regardless of infective doses of exposure, thereby reducing viral transmission. These findings will expand biopolymers’ current applicability while guiding us toward the adoption of green and eco-friendly technologies, thus reducing waste production.

## 1. Introduction

Viruses and bacteria can adhere to a wide range of surfaces and spread through similar mechanisms. One of the most clinically relevant Gram-positive bacteria is *Staphylococcus aureus*, which can rapidly form biofilms on inert surfaces, facilitating proliferation and impairing both host immune responses and the efficacy of antibiotics [1]. Other highly pathogenic microorganisms include *Enterococcus* spp., *Escherichia coli*, and *Klebsiella* spp., which are particularly associated with infections in urinary tract devices, prosthetic joints, and intravascular devices [2]. In a recent review article, Porter et al. systematically searched the Ovid MEDLINE, CINAHL, and Scopus databases for studies that described the survival time of common nosocomial pathogens in the environment. Pathogens included in the review were bacterial, viral, and fungal. They demonstrated that common pathogens of concern to infection prevention and control can survive or persist on inanimate surfaces for months [3].

Viral infections remain a persistent public health concern, often resulting in substantial medical and economic burdens. Furthermore, there are viral emergencies and re-emergencies for which there are not currently effective vaccines, antiviral treatments, or preventive strategies, underscoring the urgent need for tools that can mitigate their spread. Although direct person-to-person transmission remains the primary route for viruses such as SARS-CoV-2, indirect transmission via contact with contaminated surfaces has also been documented for this and other viruses [4,5]. Surfaces with a constant viral load, especially in high-traffic areas, can induce infections, with extended effects due to high exposure to a large number of people. For instance, experimental studies have shown that SARS-CoV-2 can survive on various materials for extended periods: up to three days on plastic and stainless steel, 24 h on cardboard, and four hours on copper surfaces [6]. Moreover, the virus was detectable in droplets suspended in the air for up to three hours. Thus, the antiviral surface treatments can help to reduce or even avoid viral transmission. These data support the need for a risk-based approach to cleaning and disinfection practices, accompanied by appropriate training, audit, and feedback, which are proven to be effective when adopted in a ‘bundle’ approach [3].

Consequently, having instruments able to stop the propagation of microorganisms is a crucial issue. Currently, this poses a significant challenge because many innovations aimed at replacing everyday items are emerging, such as metal-based hospital devices [3], antimicrobial paints [3,4,5], and replaceable surgical instruments [6], among others. However, these strategies often involve significant financial investment and are not always feasible, particularly when replacing everyday items or furnishings. A more practical and cost-effective approach is to functionalize existing surfaces using coatings capable of inactivating microorganisms, regardless of usage frequency or repeated exposure. This would allow antimicrobial protection even in domestic, educational, recreative, and other crowded centers, surpassing the ambit of the healthcare environment. This has led to growing interest in the development of antiviral/antimicrobial surface treatments [7,8].

Chirkov has highlighted a range of studied data to indicate the effectiveness of chitosan (CH) and a number of its derivatives against various kinds of viruses, despite the fact that CH is less widely known as an antiviral material [9]. Its efficacy against various viral strains can be significantly altered by replacement of the amino and hydroxyl groups [10]. To enhance its activity, antimicrobial agents such as metal nanoparticles, quaternary ammonium compounds, or antibiotics have been incorporated into coatings, either through electrostatic or covalent interactions. These coatings can function as passive pathogen barriers or active biocidal surfaces. However, their antiviral potential has been less extensively studied [11].

Copper is favored over the other metals used due to its low cost, lack of toxicity, and bioavailability [12]. In previous works, we developed CH-based formulations using citric acid (CA) as a crosslinking agent and copper salts, inducing the in situ formation of CuO NPs in the biopolymer matrix (CH.CA@Cu). This formulation demonstrated improved antiviral activity. In one such study, we probed the efficiency of CH.CA@Cu for impregnation of fabrics [13]. In addition, this formulation showed potent virucidal activity against bovine coronavirus (BCoV), herpes simplex virus type 1 (HSV-1), and bovine herpesvirus type 1 (BoHV-1).

Several challenges arise when proposing a compound with covering and antimicrobial capacity; for example, very few adhere to surfaces for long periods of time, and some alter the original appearance of the element to be covered or induce toxicity [14]. According to reported studies, few microbicide formulations suggested for coating surfaces are transparent [15,16,17]. This fact makes them unappealing for use because they alter the original object’s appearance. For example, silver-based formulations often result in a characteristic platinum hue, as observed in our previous work [18]. In addition, the coating compound must be mechanically strong enough to coat any surface under any climatic circumstances; in addition, a developed antimicrobial coating should be able to quickly inactivate any virus without endangering consumers [19]. Unfortunately, when reviewing the existent literature, it can be seen that antiviral surfaces and coatings receive comparatively less attention, with most prior studies predominantly emphasizing surface coatings endowed with antibacterial properties [20].

Although natural polymers have been explored as coating agents, sprayable formulations based on chitosan, citric acid, and copper had not been studied until our work. Here, we analyze the application of a CH.CA@Cu-based spray on different surfaces in order to gain insights into the efficiency of this biopolymeric formulation as a versatile tool for prevention of infection in community spaces. In this regard, glass, paper, plastic, and metal were chosen for the analysis as representative of surfaces usually present in high-traffic environments. We determined the antiviral activity after daily use of film-coated surfaces or repeated inoculum, after the application of CH.CA@Cu-based spray. The virucidal activity of film was evaluated over time, up to 50 days, depending on the treated surface. Furthermore, the formulation was tested against a broad range of viral types. Additionally, the biocompatibility of the dried film coating was assayed by performing skin irritation assays in rabbits. This formulation may serve as a valuable tool for surface treatment, particularly in high-traffic areas, thereby helping to reduce the spread of viruses and the risk of infection.

## 2. Materials and Methods

### 2.1. Materials

Chitosan (CH) was purchased from Farmacia Homeopática Pereda (Mar del Plata, Argentina). Citric acid was from Anedra (Buenos Aires, Argentina). Copper sulfate pentahydrate (CuSO_4_-5H_2_O) was purchased from Cicarelli (Buenos Aires, Argentina). CH.CA@Cu biopolymer solution was prepared following the procedure previously described [13]. In short, CH was combined with a 10% *w*/*v* aqueous solution of citric acid (CA) to obtain a final concentration of 8% *w*/*v*. Next, 0.5% *w*/*v* CuSO_4_ solution was added to the CH.CA solution, under vigorous stirring for 30 s, and the preparation was left to rest for 1 h. The resulting formulation was kept at 4 °C before use. A complete structural and physicochemical characterization of CH.CA@Cu was included in our previous work [13].

### 2.2. Spraying Assays

For all assays, the surfaces were cleaned with a cloth to remove loose dirt and then sprayed with CH.CA@Cu suspensions, depositing approximately 100 µL/cm^2^ at room temperature and ambient conditions. When the biopolymer suspension dried in air, it turned into thin films.

### 2.3. Scanning Electron Microscopy (SEM) Characterization

Different surfaces coated with CH.CA@Cu films were analyzed by SEM, using a FESEM-ZEISS Crossbeam 350 instrument. The micrographs were used to observe the film formation over different substrates (metal, glass, paper, and polyethylene terephthalate (PET)). For the observation, substrates were cleaned with ethanol and dried in an oven, then the solution was sprayed over the substrates and dried for 2 h at room temperature (20 °C). In addition, to examine the durability of the film, the surfaces were touched and rubbed with fingers 30–40 times (with gloves) to simulate the common uses of the surface.

### 2.4. Cell Cultures

Vero cells from African green monkey kidney cells (ATCC, CCL-81) were grown in Dulbecco’s Modified Eagle’s medium (DMEM, GIBCO); A4549 cells from human adenocarcinoma alveolar basal epithelium (ATCC, CCL-185) and HRT-18 cells derived from human large intestine adenocarcinoma (ATCC, CRL-3609) were grown in F12-MEM supplemented with 5% fetal bovine serum and were incubated at 37 °C in a 5% CO_2_ atmosphere. For maintenance conditions, the serum concentration was reduced to 1.5%.

### 2.5. Virus Stocks

The experiments were performed using human herpes simplex virus type 1 (HSV-1) strains Kos and the thymidine-kinase-deficient (tk^−^) acyclovir-resistant HSV-1 strain B2006, human herpes simplex virus type 2 (HSV-2) strain G, beta coronavirus bovine (BCoV) strain Mebus, Zika virus (ZIKV) strain Puerto Rico PRVABC59, human respiratory syncytial virus (RSV) strain A2, human adenovirus type 5 (ADV-5), and poliovirus type 1 (PV-1) vaccine strain. Viral stocks of HSV, RSV, and PV were propagated and quantified in Vero cells, BCoV was propagated in HRT-18 cells, while A549 cells were used for ADV-5 stock. For this, cells were infected and incubated at 37 °C in a 5% CO_2_ atmosphere, until the monolayer presented a predominant cytopathic effect, using an inverted microscope. The cultures were lysed by two freeze–thaw cycles, aliquoted, and stored at −80 °C. Viral titers were determined in confluent cells by the PFU technique.

### 2.6. Virucidal Activity of Biopolymer

HSV-1 viral suspensions of 20 µL, containing 10^3^ PFU, were mixed with the same volume of the CH.CA@Cu biopolymer. After 10 min, 60 µL of NaCl (2 M) was added to desorb the virion from the film, as previously reported [13,18]. We previously determined that 2 M NaCl does not affect viral infectivity [13]. For viable viral controls, the same procedure as the treatments was used by mixing 20 µL of viral suspension with 20 µL of DMEM. After centrifugation, the remainder infectivity in the supernatant was evaluated by plaque assay.

### 2.7. Virucidal Activity of Film-Coated Surfaces

#### 2.7.1. Virucidal Activity of Several Film-Coated Surfaces

Several surfaces, both film-coated and non-coated, were either infected or not with 20 μL/cm^2^, containing 10^3^ plaque-forming units (PFU) in simulated wet-droplet contamination, as described before [13], and were incubated for different times at room temperature (25 °C). Then, 80 μL of 2 M NaCl was added for every 10^3^ PFU to desorb the virion from the surfaces. After 15 min of mixing, each sample was centrifuged for 1 min at 16,000 rpm. Subsequently, the HSV-1 in the supernatants was quantified by plaque assay as described before [13,18].

#### 2.7.2. Kinetics of HSV-1 Inactivation in Films

Uncoated and film-coated PET was infected with 20 μL/cm^2^, containing 10^3^ plaque-forming units (PFU). After different times (0, 1, 15, and 30 min), 2 M NaCl was added, and the HSV-1 viral titer was quantified by plaque assay as described in Section 2.7.1.

#### 2.7.3. Virucidal Activity of Film-Coated Surfaces After Repeated Inoculations

Film-coated PET was inoculated with HSV-1 four times over 2 h, with 20 µL containing 10^3^ PFU, or four times over 24 h (every 6 h), with 20 µL containing 10^5^ PFU. Each drop was deposited on 1 cm^2^ of surface. Then, the viral titer was evaluated as described in Section 2.7.1.

#### 2.7.4. Virucidal Film Performance in Crowded Areas

Several surfaces were sprayed, and after film formation, they were subjected to daily use for 8–40 h. Afterwards, the films were scraped with a Teflon spatula. The recovered films were infected in simulated wet-droplet contamination, and the viral titer was evaluated as in Section 2.7.1.

### 2.8. Assessment of Antibacterial Activity of Formulation

*Staphylococcus aureus* (ATCC 25923, Gram-positive) and *Escherichia coli* (ATCC 25922, Gram-negative) bacteria were cultivated and maintained in nutrient broth medium. To evaluate the antibacterial activity of the CH.CA@Cu biopolymer, 1 × 10^6^ CFU (colony-forming unit)/mL of bacterial cultures were incubated at 37 °C for 24 h in the presence of varying concentrations of the biopolymer. Bacterial growth was quantified by measuring the optical density at 600 nm (OD_600_) using a spectrophotometer. Cultures without CH.CA@Cu served as negative controls. Each experimental condition was tested in duplicate.

### 2.9. Film Biosafety Analysis

#### 2.9.1. *In Vitro* Biocompatibility Evaluation

Biocompatibility of the solid films was qualitatively and quantitatively evaluated through cytotoxicity estimation by indirect exposure of Vero cells to the film extracts, as indicated in the ISO 10993-5:2009 guideline “Biological evaluation of medical devices—Part 5: Tests for in vitro cytotoxicity” [21]. Briefly, thin films were obtained by spraying 5 mL of biopolymer solution on watch glasses. After drying, the solid films were sterilized by UV irradiation for 30 min. Then, film extracts were obtained by adding 2 × 2 mm sections to 5 mL of DMEM 10% SFB for 24 h at 37 °C. For treatments, extracts were diluted at concentrations ranging from 25% to 100%. The cytotoxic effects of CH.CA@Cu film extracts were qualitatively assessed by optical microscopy, focusing on changes in general morphology, vacuolization, cell detachment, lysis, and membrane integrity. Deviations from normal morphology were documented both descriptively and numerically, in accordance with the ISO 10993-5:2009 standard (Table A1, Appendix B). For this assay, 5 × 10^5^ cells per well were seeded in a 24-well plate. After 24 h of incubation with 100% (*v*/*v*) film extract, the cells were examined using an inverted bright-field microscope, and representative images were captured. Quantitative cytotoxicity analysis was performed by estimation of cell viability through the neutral red uptake test, as we described previously [18]. To this end, 1 × 10^5^ Vero cells/well were seeded in 96-well plates in quadruplicate. Then, the cells were treated with the medium containing the film extract. The treatment with DMEM 10% SFB was considered as the control condition, while 1% hydrogen peroxide was used as a positive control for cell death. The absorbance was read at 540 nm in a Biotek Synergy HT plate reader and related to the control condition.

#### 2.9.2. Estimation of Primary Dermal Irritation Index

In vivo studies were performed in accordance with the ARRIVE guidelines in a certified laboratory (laboratory authorized by the Argentinian Ministry of Public Health by resolutions 2900–47228—Disp. 0530; 001649—Ley 11634—1443/2000; Disp. ANMAT Nro. 5892/15), following the methodology described in our previous work [18]. The assays were carried out in three healthy adult albino rabbits (marked as n° 703 (3.0 kg weight), n° 711 (2.4 kg weight), and n° 718 (2.6 kg weight)) following the OECD 404 methodology. All animals were housed individually at 21 °C, 69% humidity, and under a 12 h artificial light–12 h dark cycle, and they were fed with a conventional laboratory diet and an unrestricted supply of water. Acute dermal irritation was estimated by direct exposition of the rabbit skin to the solid film in a shaved area on the right flank. In each animal, a shaved area in the opposite flank served as a control and was treated with 0.5 mL of physiological solution. After the application, both areas were covered with sterile gauze. Erythema and edema readings were taken at 0, 24, 48, and 72 h. The Primary Dermal Irritation Index (PDII) was calculated as the resulting average scoring according to the Draize test values, as follows: 0–0.5: not irritating; 0.5–1: practically non-irritating; 1–2: minimally irritating; 2–6: moderately irritating; more than 6: severely irritating. Finally, the product was classified [22].

### 2.10. Statistical Analysis

GraphPad Prism version 7 software (San Diego, CA, USA) was used for statistical analysis. The results represent the average of at least three experiments, each condition ± SD. Statistical significance was determined by ANOVA, and *p* < 0.05 was considered significant.

## 3. Results

### 3.1. Film CH.CA@Cu Coating Performance on Different Surfaces

In our previous work [16], we demonstrated that the CH.CA@Cu formulation possesses suitable properties for impregnation of cotton fabrics, maintaining its virucidal activity even after 10 washing cycles. In this study, we aimed to evaluate the feasibility and versatility of CH.CA@Cu as a sprayable, long-lasting antiviral coating and to assess its antibacterial activity. This comprises the study of different kinds of surfaces.

Ultrathin films were obtained on a transparent Petri dish to examine the film-forming capability of CH.CA@Cu and its appearance. After the spraying and drying process, the resulting films were transparent and did not significantly affect visual clarity compared to uncoated dishes, indicating their potential for practical applications (Appendix A).

The construction process of spraying protective coatings on the surface of concrete structures is prone to defects such as uneven thickness, porosity, and inclusions. These defects can affect the protective performance of the coating to varying degrees and can even lead to coating protection failure [23]. A variety of surface materials were selected as representative of those commonly found in high-transit environments and prone to frequent contamination. To this end, glass, metal, paper, and PET surfaces were sprayed with the formulation, and their appearance and surface coverage capacity were subsequently evaluated. SEM micrographs of the coated substrates are shown in Figure 1. In all cases, the films copy the substrate surface. In the case of the metallic surface, the film peels off at the edges, whereas the rest do not show this behavior. We further studied the persistence of the films in a durability test (across the range of 30–40 touch interactions). The micrographs obtained after the experience revealed no significant changes in terms of the morphology or continuity of the formed film with respect to uncoated surfaces. In the images showing the transversal area, the film appeared fully integrated with the substrate and was indistinguishable from it, making it difficult to accurately measure the film thickness from these images. The SEM results further indicate the disappearance of cracks, wrinkles, and phase separation, thus resulting in smooth and dense coatings [24].

### 3.2. Virucidal Activity Spectrum of CH.CA@Cu Biopolymer Solution

The inhibitory effects of the biopolymer formulations can vary depending on the structural characteristics and nature of the target viruses, as well as physicochemical properties of the polymers and nanocomposites associated [25]. Therefore, we initially evaluated the virucidal activity of CH.CA@Cu biopolymer solution against a panel of viruses, representative of different structural classes and genomic types. To this end, we selected viruses known to cause various infectious diseases as well as reflecting viral diversity. Among the enveloped DNA viruses from the *Herpesviridae* family, we tested HSV-1, HSV-2, and an acyclovir-resistant HSV-1 (tk^−^) mutant strain. We also included enveloped viruses with RNA genomes like RSV, BCoV, and ZIKV, as well as the non-enveloped viruses ADV-5 and PV-1, with DNA and RNA genomes, respectively. A number of these viruses are implicated in intra infections caused in healthcare environments [26] and are responsible for a wide variety of pathologies. Regardless of their transmission route, whether or not through fomites is the main route, the viruses here studied (herpesviruses, BCoV, RSV, ADV, and PV) share the ability to transmit through this mechanism [27,28,29,30], and they can persist on surfaces for hours, as ZIKV does [31]. As shown in Figure 2, the inactivating effect of CH.CA@Cu was significantly superior, close to 100%, against enveloped viruses, while the action against non-enveloped viruses was significantly lower.

### 3.3. Persistence of Virucidal Activity on Spray-Coated Surfaces

One of the major challenges in the development of antiviral surface coatings is achieving long-lasting, reusable systems capable of continuously reducing the risk of infection and transmission [14]. To achieve this objective and the following ones, each surface was inoculated with the enveloped virus HSV-1, selected as a representative model due to the comparable inhibitory response observed among the enveloped viruses tested (Figure 3), whose incubation time is shorter than the other enveloped viruses, such as BCoV, ZIKV, or RSV (Appendix A), and because the greatest antiviral effect of CH.CA@Cu is on enveloped viruses.

To evaluate the durability of the antiviral effect of spray-coated PET surfaces, we analyzed viral titers over time following treatment with the polymeric formulation. Polyethylene terephthalate (PET) is considered an inert surface, since it does not react easily with other materials [32]. Therefore, to evaluate the antiviral effects of CH.CA@Cu films, we initially chose PET as the surface to be sprayed. As shown in Figure 3A, the coated films exhibited excellent virucidal activity against an inoculum of 10^3^ PFU. Additionally, Figure 3B shows the analysis of the persistence assay using HSV-1 on film-coated PET surfaces. These results demonstrate that the inactivation of HSV-1 is maintained for up to 50 days without a significant reduction in efficacy.

### 3.4. Inactivating Performance After Repeated Inoculation

Since pathogens can persist on contaminated surfaces for long periods of time and these surfaces can be re-exposed to reinoculation [33], we proposed the following experiments. In order to explore the antiviral performance of the film coatings, the subsequent studies pointed to determining its deleterious capacity when the viral inoculum was increased in time in two settled schemes. Film-coated surfaces were exposed to four inoculations of 10^3^ PFU over 2 h, or four times over 24 h with 10^5^ PFU, thus representing a more intense virus exposure. As shown in Figure 3C, in both conditions, the CH.CA@Cu-coated PET surfaces achieved complete inactivation of the HSV-1 load, with a 100% reduction in viral titer.

### 3.5. Virucidal Activity of Films Applied to Different Surfaces

Given that the primary goal of developing these films is to coat surfaces that inherently lack antiviral properties, we further evaluated the virucidal efficacy of CH.CA@Cu films after application to various substrates, including PET, stainless steel, nickel-plated steel, aluminum, wood, leather, cotton fabric, and glass. As shown in Table 1, the film-coated surfaces exhibited rapid virucidal activity, achieving at least 90% inactivation of HSV-1 within the first minute of contact, and this effect was further enhanced after 30 min in all film-coated surfaces, with the exception of aluminum. In this case, the reduction in viral infectivity was only 56% at 1 min, increasing slightly to 60% after 30 min.

### 3.6. Field Study of Virucidal Film Performance in Crowded Areas

The inactivating film capacity was evaluated after application to various surfaces exposed to routine use in COVID-19 care units and short-distance public buses. The recovery of naturally occurring SARS-CoV-2 infectivity was not studied in this trial. The sprayed surfaces are detailed in Table 2. After application, surfaces were subjected to daily use during indicated times, after which the films were recovered, and their virucidal activity was evaluated.

Given the pace of work in the health primary care units, access for sampling was granted at 16 and 40 h post-spray. In the public transportation sector, sampling was permitted only at 8 h post-spray. We observed a conservative anti-HSV-1 activity of films after all tested exposure times, demonstrating persistent virucidal capacity under real-use conditions.

### 3.7. Antibacterial Activity

The antibacterial activity of the CH.CA@Cu was evaluated by measuring the inhibition of growth of Gram-negative (*E. coli*) and Gram-positive (*S. aureus*) bacteria. As shown in Figure 4, bacterial exposure to increasing concentrations of CH.CA@Cu resulted in a significant reduction in OD_600_ values at concentrations of 0.1% *v*/*v* or higher. Specifically, treatment with 0.1% CH.CA@Cu reduced bacterial growth by over 60%, while a 5% concentration further decreased activity by approximately 10% compared to the control.

### 3.8. In Vitro Biocompatibility Evaluation

To evaluate the biocompatibility of the film composition in vitro, we employed a strategy previously used in our earlier studies [18]. Since the films dissolve upon contact with aqueous solutions, we followed the protocol in the ISO 10993-5:2009 guideline [21]. Vero cells were incubated for 24 h at 37 °C with the complete dissolution product of the films, hereafter referred to as the ‘film extract’, which contains all the soluble and insoluble products of the film in contact with aqueous media.

Following the ISO standard, both qualitative and quantitative analyses of cytotoxicity, as a direct indicator of biocompatibility, were performed. The qualitative analysis involved light microscopic examination of cell morphology. As shown in Figure 5A, the exposure of confluent Vero cell culture to the undiluted (100%) film extract for 24 h did not alter the cellular morphology compared to the control. No signs of cytolysis, discrete intracytoplasmic granules, cellular lysis, or inhibition of proliferation were observed. According to this result, and considering the scoring system presented in Appendix A, the film extract does not induce any cytotoxic reaction (reactivity score = 0).

On the other hand, the quantitative evaluation of cell viability was conducted using the neutral red uptake assay to determine whether the film extract exerted any toxic effect on Vero cells. As shown in Figure 5B, no significant differences in absorbance measures were observed between control cells and those exposed to different concentrations of film extracts, indicating the absence of cytotoxic effect. In contrast, the positive control (hydrogen peroxide) induced an 80% reduction in cell viability (Figure 5B) (*** *p* < 0.001). In this case, the total absence of a cytotoxic effect from the film extracts indicates that the film composition displays high biocompatibility in vitro in Vero cell cultures.

### 3.9. In Vivo Safety Analysis: Acute Dermal Irritation Index Estimation

After confirming that the material extracts are biocompatible with Vero cultured cells, the next step was to evaluate whether the solid films are biosafe for the skin in their intended application on surfaces. For this purpose, an animal model was used to assess acute primary dermal irritation, following the OECD 404 guideline. Adult rabbits were selected for the study.

As shown in Figure 5C, direct exposure of rabbit skin to the dry films for 24, 48, and 72 h did not produce any detectable erythema or edema compared to the control condition (saline solution). Therefore, the Primary Dermal Irritation Index, calculated as the mean score from three animals, was zero, classifying the material as non-irritant.

Based on the results presented in the previous two sections, CH.CA@Cu film extracts exhibit high biocompatibility and a lack of cytotoxicity. Furthermore, the solid films themselves do not show potential for acute skin irritation—evidenced by the absence of erythema or edema even after prolonged exposure.

## 4. Discussion

In our previous work, we characterized the CH.CA@Cu impregnated textiles and demonstrated their antiviral activity against herpesvirus and bovine coronavirus [13]. This work demonstrates its versatility to be sprayed on different surfaces, with the goal of forming an antimicrobial film and to expand its activity against a wider gamut of viruses. There are few studies that propose antimicrobial compounds for surface coating and evaluate the transparency and durability of the film despite the uses [8]. Herein, we demonstrated that a thin layer of CH.CA@Cu film is able to adhere and form a continuous layer on various surfaces, remaining almost unaltered despite use.

Regarding the antiviral properties achieved through this work, while CH.CA@Cu’s activity against non-enveloped viruses was much lower, its inactivating effect against enveloped viruses was noticeably better, approaching 100%. In this context, we can note that various enveloped viruses were tested, which can be transmitted through fomites (BCoV, herpesviruses, and RSV), and they are implicated in intra infections caused in healthcare environments [26]. In addition, a novel virus, ZIKV, was included, as it had not been considered for this type of study before. Even when fomite transmission is not considered its primary route of infection, it can last on surfaces for hours [31]. These results suggest that CH.CA@Cu can be used as a virucidal tool, with optimal results against enveloped viruses and satisfactory results against naked viruses. Previous studies described that CH may interact electrostatically with viral surface proteins; this type of electrostatic binding prevents the interaction of viruses with cells susceptible to infection, thereby avoiding viral attack on cells [34]. However, this mode of action does not lead to virion destruction. The virucidal activity is defined as the irreversible inactivation or destruction of viral particles. Although CH’s precise mode of action on viruses is unknown, it is known that a variety of mechanisms can alter viral infectivity; for example, it is known that given its polycationic characteristic, it binds to anionic surfaces of viruses, disrupting the viral membrane, thus destroying enveloped viruses [35]. It is hypothesized that CH can attach to the capsid of naked viruses, blocking viral adsorption like that of the HPV virus, but it does not destroy it. Thus, it is also observed that CH decreases the infectivity of naked viruses, but in some cases this effect is almost imperceptible [35]. For acetic acid, it is known that it can prevent the binding of cellular receptors to viral glycoproteins of enveloped viruses; in contrast, in naked viruses, its mode of action is unknown, but a physical inhibition is suggested [36]. In Cano-Vicent’s work, it is observed that chitosan–acetic acid formulations also have antiviral action against enveloped and naked viruses. However, against the latter, this activity tends to be lower [37].

On the other hand, the contribution of ion metals, such as copper, has been reported to have antiviral action, where the suggested mechanisms indicate the generation of ROS as responsible for virus destruction [38].

It is worth mentioning that our results show that the inactivation of HSV-1 by CH.CA@Cu is maintained for up to 50 days without a significant reduction in efficacy. This finding is encouraging given that it is known that polycation polymers lose their antiviral activity over time [14]. Our results suggest that the long-term activity may be attributed to the presence of copper in the formulation, consistent with previous observations in fabric impregnated with copper-containing coatings, which retained inactivating performance after multiple washes, unlike those lacking copper [13].

To date, information in the literature specifying exact amounts of virus per aerosol droplet from infected patients is missing. However, the amount of RNA from respiratory viruses such as influenza in aerosols has been measured at a concentration equivalent to 10–100 viral particles in a droplet [39]. Also, doses above 10^5^ particles of SARS-CoV 2 per droplet are considered exaggerated [39,40]. Considering these data, and the fact that the viral load per droplet depends on the patient’s severity, strength of expectoration, age, etc., in our study, we decided to work with an infectious viral range just above the aforementioned upper limit to ensure we covered all possible scenarios, especially the most infectious one in a simulated state of maximum expected infectivity. Thus, the initial study dose was 10^3^ PFU per drop, which is consistent with that used by Warnes et al. [41] and also our own previous works [13,18].

With the exception of aluminum, all film-coated surfaces demonstrated rapid virucidal activity, as demonstrated in Table 1, with at least 90% of the HSV-1 inactivated within the first minute of contact. In the aluminum case, the reduction in the virucidal activity of the film can be related to the strong interactions between CH and Al moieties. Aguilar-Ruiz et. al, in their study which aimed to develop a crosslinked chitosan-based coating from shrimp waste as an alternative to expensive commercial coatings, demonstrated that chitosan-based coatings showed promise as effective corrosion inhibitors. They found the occurrence of molecular interactions and structural alterations of biopolymer’s structure according to FTIR-ATR analysis [42]. These robust interactions strongly comprised the CH functional groups implicated in the inactivation of the viruses, reducing the efficiency of the formulation.

One important issue when proposing an antiviral formulation is its efficiency after repeated inoculations. There is scarce literature presenting studies where the antiviral compound coated surface capacity is evaluated against repeated inoculations. In the contribution of Urmi et al., cationic compounds coated surfaces present high antiviral activity against the first viral inoculation, but after following inoculations this efficiency decreases [43]. However, our results indicate that CH.CA@Cu-coated PET surfaces maintain their antiviral potential despite being repeatedly inoculated with viruses (Figure 3C). Therefore, we consider that the results presented here are promising and have great potential to be used in the prevention of transmission of infections by fomites even after being exposed to repeated viral inoculations.

Our CH.CA@Cu biopolymer also proved to be effective against bacterial infections. CH.copper nanocomposites have been previously tested in terms of their antibacterial effects against *S. aureus* [44], but their effect on *E. coli* had not been reported prior to this study. Several mechanisms have been proposed for the antibacterial action of CH and its derivatives, including disruption of the bacterial membrane and damage to cellular components, ultimately leading to microbial cell death [45]. The reported infectious bacterial doses widely vary and depend on the pathogen being evaluated. For example, data indicate that in operating rooms, the bacterial colony load should not exceed 180 CFU/m^3^ [46]; for Salmonella, the infective doses are around 10^6^–10^8^ CFU [47]; adults can be infected with doses of 10^2^ CFU of Shigella [43]; doses of enteropathogenic *E. coli* bacteria lower than 10^8^ cause serious harm to individuals [48]; and cholera infections are acquired with doses of 10^4^ [49]. Therefore, the doses used in this study appear as a representative range of bacterial infective doses. This indicates that the formulations proposed here could be efficient to mitigate both viral and bacterial infections caused by the most common infective doses.

According to biosafety requirements, the achieved evaluation data reveal that the tested films are non-cytotoxic to Vero cells and non-irritant in rabbit skin, thus supporting the safe application of this nanocomposite in biological environments. In contrast, CH–silver nanocomposite films, while demonstrating antibacterial and antiviral effects, have been shown to compromise cytocompatibility due to metal ion release [50]. In this sense, the current literature on CH nanoparticles highlighted that even if the CH-nanosystem displays low cytotoxicity across various formulations and cell lines, it is mandatory to empathize the need for safety assessment for each derivative [51].

## 5. Conclusions

The CH.CA@Cu sprayable formulation can be easily applied to existing surfaces without the need to replace any materials. This not only reduces costs and time but also offers a more sustainable alternative by minimizing waste generation.

Its demonstrated virucidal and antibacterial properties contribute to reducing the transmission of infections agents in crowded public environments, such as hospitals, schools, and public transportation. This will not only protect the health of individuals using these facilities but also contribute to decreasing the overall spread of pathogens in the broader community. In addition to its antimicrobial properties, this sprayable formulation improves the durability of treated surfaces, making them particularly suitable for high-traffic areas where hygiene and resistance to wear are essential. From this evidence, it could be proposed as a promissory solution for high-traffic areas where hygiene is a top priority.

In summary, surface coating with CH.CA@Cu represents a practical, cost-effective, and environmentally friendly approach for enhancing biosecurity in shared public spaces. Coating surfaces in public facilities is an effective and sustainable solution for preventing the spread of microbial infections. Its affordability, ease of application, and multiple benefits make it a promising candidate for widespread real-world implementation.

## Figures and Tables

**Figure 1 viruses-17-01347-f001:**
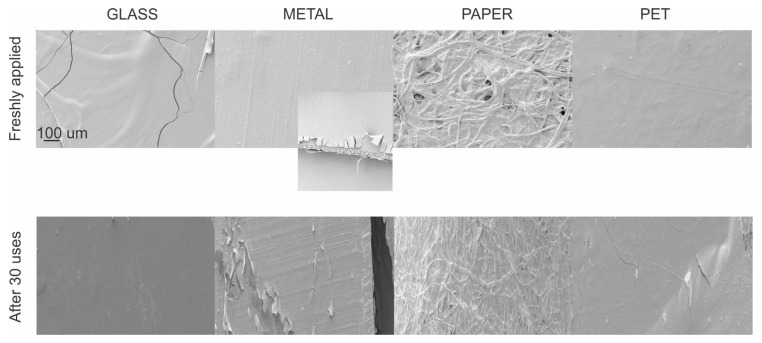
SEM images of coated substrates, including glass, metal, paper, and PET.

**Figure 2 viruses-17-01347-f002:**
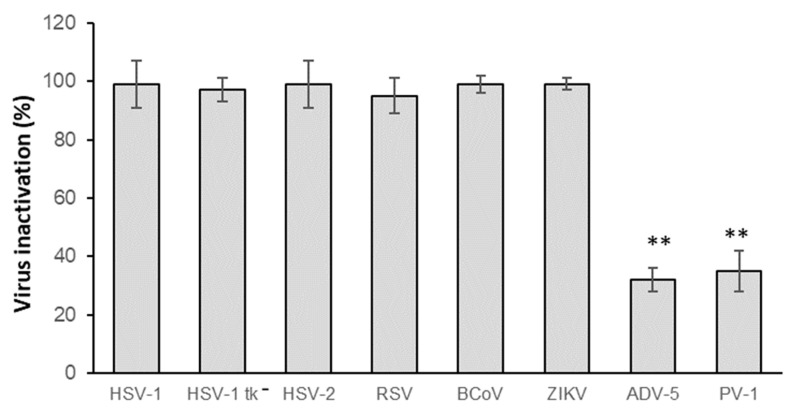
Virucidal activity of CH.CA@Cu biopolymer. The mean ± SD of the data from a minimum of three different experiments is used to express the results. Each condition was processed in duplicate. ** *p* < 0.01 vs. HSV-1 condition.

**Figure 3 viruses-17-01347-f003:**
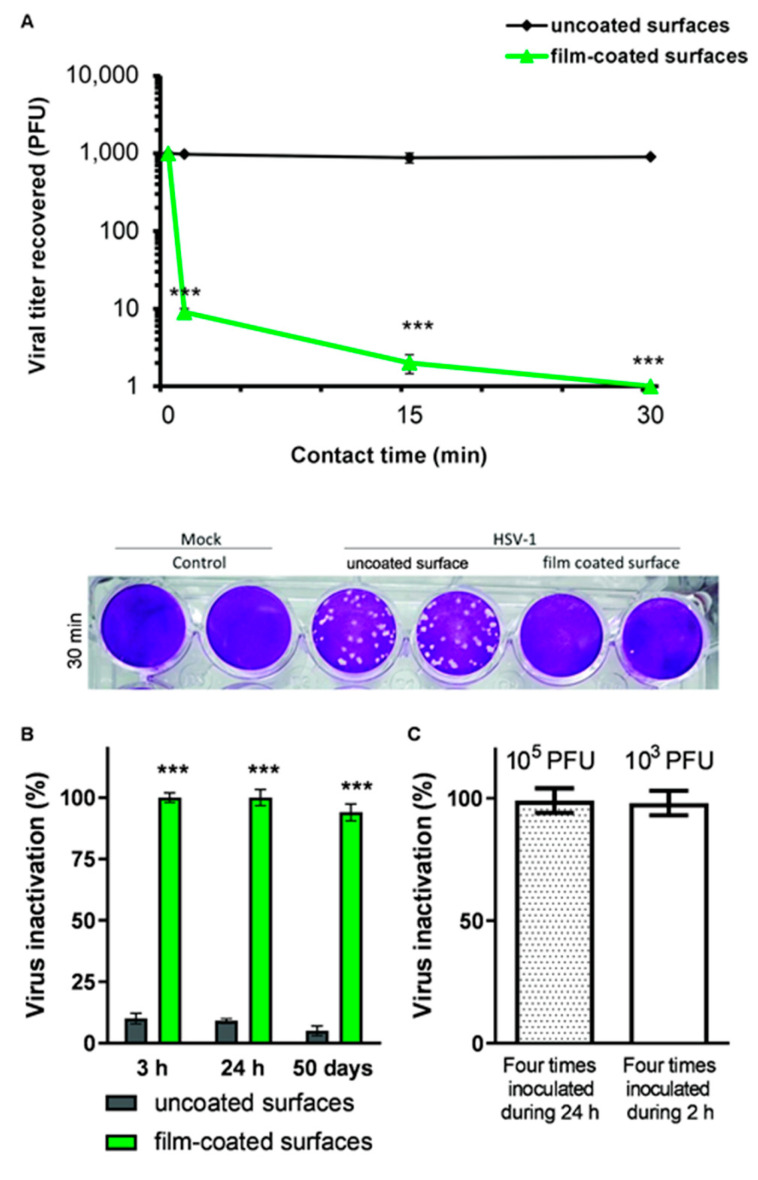
(**A**) Top: HSV-1 titer recovered after different contact times with film-coated or uncoated PET surfaces. Bottom: Representative image of plaque formation corresponding to the top figure. (**B**) Persistence of virucidal properties of coated surfaces after different times after the polymeric formulation was sprayed. (**C**) Assessment of virus titers on film-coated surfaces after repeated viral inoculations. Data from three experiments are shown, with error bars denoting ± SD *** *p* < 0.001 vs. uncoated surface.

**Figure 4 viruses-17-01347-f004:**
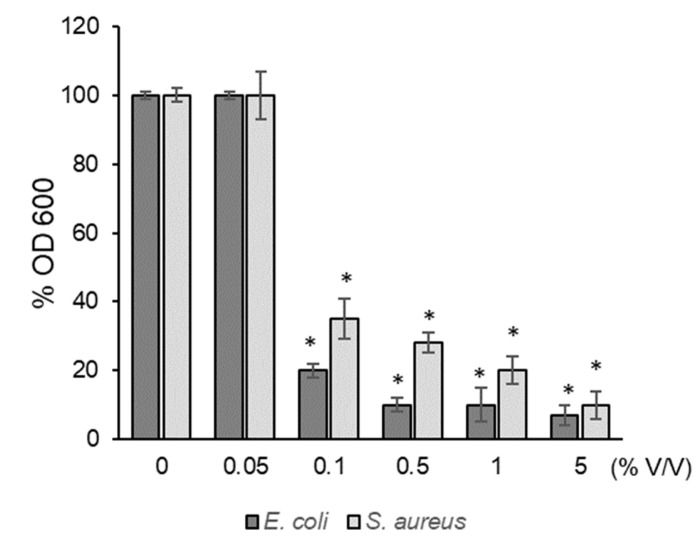
Antibacterial activity of CH.CA@Cu on *E. coli* and *S. aureus*. Means and SD are shown. * *p* < 0.05 with respect to the control.

**Figure 5 viruses-17-01347-f005:**
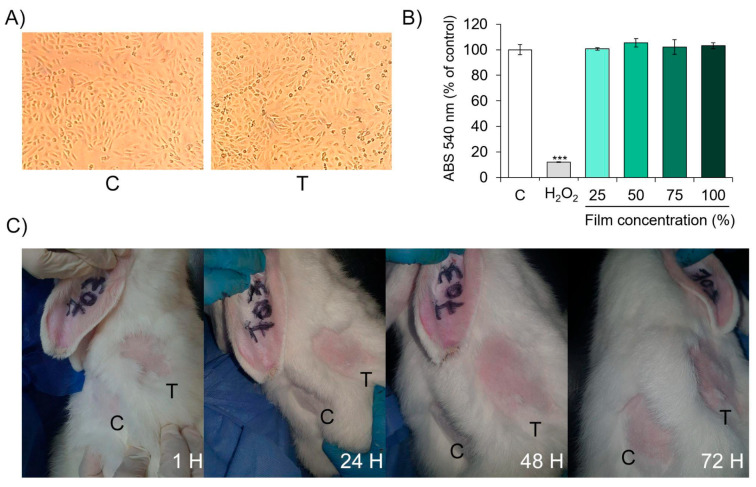
Biosafety analysis of the CH.CA@Cu biopolymer. (**A**,**B**) In vitro assessment of biocompatibility. (**A**) Qualitative analysis of cell integrity. Vero cells were exposed for 24 h to 100% concentrated extracted medium. Representative images show cell morphology and integrity. Control (C) and treated (T) conditions. (**B**) In vitro quantitative analysis of the impact of CH.CA@Cu biopolymer on cell viability. Vero cell cultures were exposed for 24 h to increasing concentrations of the film-extracted medium (25 to 100% concentration) or to 1% hydrogen peroxide. Cell viability was assessed by NRU assay. Absorbance (ABS) values of two independent experiments performed in quadruplicate were normalized to the untreated control cells and expressed as a percentage. Means and SD are shown. *** *p* < 0.001 with respect to the control. (**C**) Estimation of Primary Dermal Irritation Index. Albino rabbits were directly exposed to CH.CA@Cu biopolymer solid film in a shaved area for 1 to 72 h. In the same animal, another shaved area was used as C and T. Representative images of the treated areas across the exposure times are shown.

**Table 1 viruses-17-01347-t001:** Effect of exposure time on HSV-1 titers on treated surfaces.

	HSV-1 Titer Reduction [%]	
Treated Material	1 min	30 min
PET	99.5 ± 0.7	99.9 ± 0.9
Stainless steel	95.1 ± 0.3	99.0 ± 0.1
Nickel-plated steel	99.9 ± 0.1	100.0 ± 0.2
Aluminum	56.0 ± 0.6	60.0 ± 0.1
Wood	99.6 ± 1.1	100.1 ± 0.1
Leather	99.8 ± 0.1	99.8 ± 1.1
Cotton fabric	90.1 ± 3.1	95.0 ± 0.2
Glass	98.1 ± 1.1	99.6 ± 2.6

**Table 2 viruses-17-01347-t002:** HSV-1 viral titer in recovered films.

	HSV-1 Titer Reduction (%)	
Health Unit Primary Care	16 h	40 h
Eco-leather from chairs	90.9 ± 0.2	95.0 ± 2.1
Wood table reception	97.8 ± 0.5	100.2 ± 1.2
Glass window	97.2 ± 0.7	100.0 ± 0.3
Quartz countertops	99.5 ± 1.2	98.1 ± 0.5
Stretcher	98.1 ± 0.5	96.8 ± 1.8
Disposable friselina gown	92.3 ± 1.2	99.5 ± 4.2
Surgical mask	98.3 ± 2.1	95.6 ± 0.5
**Buses (public transport)**	**8 h**	
Eco-leather from chairs	99.5 ± 0.9	
Glass windows	99.8 ± 1.1	
Stainless steel railings	99.2 ± 1.9	

## Data Availability

All data is available within the manuscript.

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
