# Peer review of "Virucidal and Antibacterial Chitosan–NanoCu Film-Coating-Based Technology: Complete Analysis of Its Performance on Various Surfaces"

_viruses, 2025, doi:10.3390/v17101347_

Round 1

Reviewer 1 Report

Comments and Suggestions for Authors
  1. I would like to congratulate the authors on the excellent work they have done in this work. The authors have assessed the material's ability to exhibit strong antibacterial and antiviral properties. Particularly A. aureus and E. coli which gram negative bacteria. I would like to know why they chose only these bacteria not included some Gram-positive bacteria. 
  2. It will be recommended that they also evaluate impact of factors e.g pH, concentration, molecular weight and degree of deacetylation on the material efficancy.
  3. Is the biocompatibility and the safety of this material assessed? How?
  4. Did authors examine the physical and chemical propoerties of the nanoCu filmcoating, including its stability, solubility and its potential for chemical modification.
  5. How did you choose the viruses that were used for antiviral testing?

Author Response

Reviewer #1

I would like to congratulate the authors on the excellent work they have done in this work. The authors have assessed the material's ability to exhibit strong antibacterial and antiviral properties.

  1. Particularly A. aureus and E. coli which gram negative bacteria. I would like to know why they chose only these bacteria not included some Gram-positive bacteria. 

Author’s response: Thank you for your comment, but in our work, we used both Gram-positive and Gram-negative bacteria, which was mentioned in the following.

See lines 193:

“2.8. Assessment of antibacterial activity of formulation

Staphylococcus aureus (ATCC 25923, Gram-positive) and Escherichia coli (ATCC 25922, Gram-negative)”

See lines 352:

The antibacterial activity of the CH.CA@Cu was evaluated by measuring the inhibition of growth of Gram-negative (E. coli) and Gram-positive (S. aureus) bacteria.”

  1. It will be recommended that they also evaluate impact of factors e.g pH, concentration, molecular weight and degree of deacetylation on the material efficancy.

Author’s response: The query of the Reviewer is of concern. We have deeply studied the structural and physicochemical properties of chitosan-loaded Cu and Ag NPs in situ formed. Please revise the following publications:

- Development and characterization of antimicrobial textiles from chitosan-based compounds: Possible biomaterials against SARS-CoV-2 viruses, María Florencia Favatela; Jessica Otarola; Victoria Belen Ayala-Peña; Guillermina Dolcini; Sandra Perez; Andrés Torres Nicolini; Vera Alejandra Alvarez; Verónica Leticia Lassalle, Journal of Inorganic and Organometallic Polymers and Materials. https://doi.org/10.1007/s10904-021-02192-x. 12/2021.

- Chitosan-based formulations intended as masks surface protective spray for prevention of coronavirus dissemination Victoria Belen Ayala-Peña, María Julia Martin, Florencia Favatela, Jessica Otarola, María Ventura, Claudia Gentili, María Florencia Salcedo, Andrea Mansilla, Sandra Pérez, Guillermina Dolcini, Vera Alvarez, and Verónica Lassalle, Chemistry Select, ChemistrySelect 2022, 7(37):e202202410. doi: 10.1002/slct.202202410.

From the data achieved in previous contributions, we optimize the formulation of the film included in the present work. In the case of the pH we have not possibilities to modify it because of the stability of biopolymeric moieties. It is well known that pHs higher than 4 -4.5 lead he precipitation of chitosan, hence the application as spray on different surfaces would be impossible.

Besides we have solid experience in the design and preparation of chitosan based biomaterials, nanosystems,etc. so it is true, as the Reviewer suggest, that the acetylation degree and molecular weight of biopolymer may have an impact on the properties of chitosan. However, exploring these issues related to the biopolymer requires a deep study that clearly remained out of the scope of the present contribution, where the protective properties against different surfaces is the focus. The comment of the reviewer can be considered for a future separated article.

  1. Is the biocompatibility and the safety of this material assessed? How?

Author’s Response: Yes, we have analyzed the material's biocompatibility and biosafety through both in vitro and in vivo studies. These results are detailed in Sections 3.8 (In vitro cytotoxicity evaluation) and 3.9 (Acute dermal irritation analysis), respectively.

Specifically, Section 3.8 presents data from an assay conducted in accordance with the ISO 10993-5:2009 guideline for the cytotoxicity evaluation of medical devices. The assay involved exposing cell cultures in the exponential growth phase to increasing concentrations of a film extract. This extract was obtained by incubating dried, thick sections of the film in culture medium for 24 hours at 37°C to determine whether any soluble components could affect cell viability in vitro. We observed that the viability and morphology of Vero cell cultures were not affected after 24 hours of incubation with the 100% concentrated (undiluted) film extract. These data were assessed both qualitatively via light microscopy and quantitatively using the Neutral Red uptake assay, as described in the Methodology section.

Conversely, we conducted in vivo experiments to demonstrate that direct skin exposure to the film-coated surfaces would not cause any acute harm. These data are included in Section 3.9. This was achieved by directly exposing rabbit skin to the dry films for 24, 48, and 72 hours; this exposure did not produce any detectable erythema or edema compared to the control condition (saline solution).

Together, the results obtained from both in vitro and in vivo studies allow us to conclude that the films are biosafe for their intended use.

As the reviewer's comment indicates that this conclusion was unclear in the original manuscript, we have now modified the text in both sections as follows, see line 376:

3.8. In vitro cytotoxicity biocompatibility evaluation 

To evaluate the biocompatibility  impact of the film composition in vitro on cell cultures, we employed a strategy previously used in our earlier studies ​[21]​. Since the films dissolve upon contact with aqueous solutions, we followed the protocol in the ISO 10993-5:2009 guideline ​[31]​. Vero cells were incubated for 24 h at 37 °C with the complete dissolution product of the films, hereafter referred to as the 'film extract', which contains all the soluble and insoluble products of the film in contact with aqueous media.  

Following the ISO standard, both qualitative and quantitative analyses of cytotoxicity, as a direct indicator of biocompatibility, were performed. The qualitative analysis involved light microscopic examination of cell morphology. As shown in Figure 6A, the exposure of confluent Vero cell culture to the undiluted (100%) film extract for 24 h did not alter the cellular morphology compared to the control. No signs of cytolysis, discrete intracytoplasmic granules, cellular lysis, or inhibition of proliferation were observed. According to this result, and considering the scoring system presented in Table 1S, the film extract does not induce any cytotoxic reaction (reactivity score = 0).  

On the other hand, the quantitative evaluation of cell viability was conducted using the neutral red uptake assay to determine whether the film extract exerted any toxic effect on Vero cells. As shown in Figure 6B, no significant differences in absorbance measures were observed between control cells and those exposed to different concentrations of film extracts, indicating the absence of cytotoxic effect. In contrast the positive control (hydrogen peroxide) induced an 80% reduction in cell viability (Figure 6B) (*** p < 0.001). In this case, the total absence of a cytotoxic effect from the film extracts indicates that the film composition displays high biocompatibility in vitro in Vero cell cultures.

3.9. In vivo safety analysis: Acute dermal irritation index estimation analysis 

After confirming that the material extracts are biocompatible with Vero non-toxic to cultured cells, the next step was to evaluate whether the solid films are biosafe for the skin in their intended application on surfaces. For this purpose, an animal model was used to assess acute primary dermal irritation, following the OECD 404 guideline. Adult rabbits were selected for the study. 

As shown in Figure 6C, direct exposure of rabbit skin to the dry films for 24, 48, and 72 h did not produce any detectable erythema or edema compared to the control condition (saline solution). Therefore, the Primary Dermal Irritation Index, calculated as the mean score from three animals, was zero, classifying the material as non-irritant.

Based on the results presented in the previous two Sections, CH.CA@Cu film extracts exhibit high biocompatibility and a lack of cytotoxicity. Furthermore, the solid films themselves show no potential for acute skin irritation—evidenced by the absence of erythema or edema even after prolonged exposure

  1. Did authors examine the physical and chemical propoerties of the nanoCu filmcoating, including its stability, solubility and its potential for chemical modification.

Author’s Response: As it was mentioned in point 2, the physicochemical properties as well as stability and other related issues have been addressed in previous works. In this contribution the focus was particularly on the application of films as protective coating indifferent surface. Given the profile of the journal (Viruses) the antiviral and antibacterial properties and the perspectives of concrete application of the biopolymeric formulation are highlighted.

  1. How did you choose the viruses that were used for antiviral testing?

Author’s Response: In our work, we selected a wide range of viruses, attempting to capture a diverse spectrum in terms of viral characteristics (genome type and envelope) and the type of pathology they cause. They could be transmitted to a greater or lesser extent through fomites. We selected enveloped and DNA-type viruses (herpesviruses), enveloped and RNA-type viruses (BCoV, RSV, and ZIKV), naked and DNA-type viruses (ADV), and naked RNA-type viruses (PV-1).

Herpesviruses cause exanthematous pathologies, but in some cases can lead to herpes encephalitis. They are highly prevalent in the global population, and their transmission is primarily through direct contact and, to a lesser extent, through fomites. Therefore, proper cleaning of these types of fomites is recommended, especially if they are shared in an environment where a person with active genital or oral herpes lesions uses one of these fomites shortly after another person does so (Suissa  et al., 2023). Its complications can be serious in immunocompromised patients. The HSV-1 (tk-) virus is added because it is a strain resistant to acyclovir. This antiviral is widely used in the treatment of herpes viruses, but it is known to generate drug resistance. Therefore, we added this strain to determine if the effect of this antiviral is also potent against this type of virus.

BCoV is a virus belonging to the SARS-CoV-2 family. It causes respiratory symptoms and can be associated with systemic complications. Its second route of transmission is through fomites. It is known that it can be deposited on surfaces after the settling of respiratory droplets expelled by infected patients. (Castaño et al., 2021). It has been found in hospitals and apartments on telephones, doorknobs, computer mice, toilet handles, latex gloves, sponges (Boone et al., 2007)

RSV: It is a virus that causes severe bronchiolitis in children and the elderly and is a cause of hospitalizations. It is primarily transmitted through respiratory droplets and secretions expelled when coughing or sneezing. It is also transmitted through direct contact with an infected person (such as a kiss) or by touching contaminated surfaces or objects and then touching the mouth, nose, or eyes. It has been found in hospitals, for example on countertops, cloth gowns, rubber gloves, paper tissues, and hands (Boone et al., 2007)

ZIKV: It is a virus that causes neurological symptoms in adults and brain development disorders in the fetus. Although transmission through fomites is not the main route of transmission, it is known that this virus can persist on surfaces for hours and can be transmitted sexually (Müller et al., 2016). Furthermore, this virus was included to demonstrate that these antivirals can be used against other viruses not typically transmitted by fomites, so these compounds could be used beyond the proposed applications for surfaces. The results obtained are novel and have potential for future application in other presentations or applications.

ADV: These types of viruses cause a variety of pathologies, including respiratory, ocular, and/or gastrointestinal diseases. Infection is generally contracted through contact with respiratory secretions from an infected person or with a contaminated object (e.g., towels, instruments). Infection can be transmitted through the air or through water (e.g., contracted by swimming in lakes or pools without adequate chlorine). Asymptomatic respiratory or gastrointestinal viral shedding can continue for several months or even years. In our case, ADV-5 was chosen because it is representative of ADVs. It has been found in bars and cafes on cups, paper, porcelain, cotton fabric, latex, glazed tiles, and polystyrene (Boone et al., 2007)

Poliovirus:  The infection causes fever and stomach pain, but in some cases it can progress to muscle paralysis. It is primarily transmitted from person to person via the fecal-oral route, through the ingestion of water or food contaminated with infected feces. It can also be spread through the respiratory route, via droplets of saliva or aerosols from the throat of an infected person. Transmission occurs through direct contact with infected feces or contaminated objects, or through contact with respiratory secretions (Lopez et al., 2013).

Thus, when selecting viruses, we proposed covering the four major groups of viruses (naked-DNA genome, naked-RNA genome, enveloped-DNA genome, enveloped-RNA genome) for testing antiviral compounds against viruses that could cause a wide variety of pathologies: exanthematous, neurological, gastrointestinal, respiratory, and systemic.

Regardless of their primary transmission route, they could be transmitted to a greater or lesser extent through fomites. Now we add this last sentence to the text as follows, see lines 287:

“Therefore, we initially evaluated the virucidal activity of CH.CA@Cu biopolymer solution against a panel of viruses, representative of different structural classes and genomic types. To this end, we selected viruses known to cause various infectious diseases and reflecting viral diversity. Among the enveloped DNA viruses from the Herpesviridae family, we tested HSV-1, HSV-2, and an acyclovir-resistant HSV-1 (tk⁻) mutant strain. We also included enveloped viruses with RNA genomes like RSV, BCoV, and ZIKV, as well as the non-enveloped viruses ADV-5 and PV-1, with DNA and RNA genomes, respectively. Various of these viruses are implicated in intra infections caused in healthcare environments [27] and are responsible for a wide variety of pathologies. Regardless of their transmission route, whether through fomites is the main route or secondary, the viruses studied here (Herpesviruses, BCoV, RSV, ADV, PV) share the ability to transmit through this mechanism [26–29], or although transmission through fomites has not been reported, they can persist on surfaces for hours, as ZIKV does [30]. 

Added references

Suissa, C.A.; Upadhyay, R.; Dabney, M.D.; Mack, R.J.; Masica, D.; Margulies, B.J. Investigating the Survival of Herpes Simplex Virus on Toothbrushes and Surrogate Phallic Devices. Int J STD AIDS 2023, 34, 152–158, doi:10.1177/09564624221142380.

Castaño, N.; Cordts, S.C.; Kurosu Jalil, M.; Zhang, K.S.; Koppaka, S.; Bick, A.D.; Paul, R.; Tang, S.K.Y. Fomite Transmission, Physicochemical Origin of Virus–Surface Interactions, and Disinfection Strategies for Enveloped Viruses with Applications to SARS-CoV-2. ACS Omega 2021, 6, 6509–6527, doi:10.1021/acsomega.0c06335.

Boone, S.A.; Gerba, C.P. Significance of Fomites in the Spread of Respiratory and Enteric Viral Disease. Appl Environ Microbiol 2007, 73, 1687–1696, doi:10.1128/AEM.02051-06.

Lopez, G.U.; Gerba, C.P.; Tamimi, A.H.; Kitajima, M.; Maxwell, S.L.; Rose, J.B. Transfer Efficiency of Bacteria and Viruses from Porous and Nonporous Fomites to Fingers under Different Relative Humidity Conditions. Appl Environ Microbiol 2013, 79, 5728–5734, doi:10.1128/AEM.01030-13.

Müller, J.A.; Harms, M.; Schubert, A.; Jansen, S.; Michel, D.; Mertens, T.; Schmidt-Chanasit, J.; Münch, J. Inactivation and Environmental Stability of Zika Virus. Emerg Infect Dis 2016, 22, 1685–1687, doi:10.3201/eid2209.160664.

Some phrases were also added to the discussion. See line 434:

Regarding the antiviral properties achieved through this work, while CH.CA@Cu's activity against non-enveloped viruses was much lower, its inactivating effect against enveloped viruses was noticeably better, approaching 100%. In this context, we can note that various enveloped viruses were tested, which can be transmitted through fomites (BCoV, herpesviruses, RSV), and they are implicated in intra infections caused in healthcare environments [25]. In addition, a novel virus was included, as it had not been considered for this type of study before, ZIKV, since fomite transmission is not considered its primary route of infection, however; it can last on surfaces for hours [30]. These results suggest that CH.CA@Cu can be used as a virucidal tool, with optimal results against enveloped viruses and considerable results against naked viruses.

Reviewer 2 Report

Comments and Suggestions for Authors

This manuscript presents a highly relevant topic. The article is well-structured and the findings are compelling, demonstrating a clear progression from concept to laboratory validation. The relevance of this research is exceptionally high. The persistent problem of healthcare-associated infections transmitted via contaminated surfaces remains a significant burden on global healthcare systems, both in terms of patient outcomes and financial costs. This issue was starkly highlighted during the COVID-19 pandemic, which also led to a massive increase in single-use plastic waste. The novelty of the work is clearly articulated and lies in the multifunctional and sustainable approach to creating protective surfaces. The scientific significance of this study is substantial. It contributes to the growing field of advanced, functional materials by expanding the known applicability of chitosan-based materials beyond their traditional uses in packaging or wound dressings into the realm of surface disinfection and infection control. The practical implications of this research are profound and directly align with the stated goals infection control and economic benefits.

Comment 1: In the Introduction, the antibacterial and antiviral activities of the chitosan-nanoCu film should be delineated more clearly. The text on lines 43-59 first covers antibacterial activity, is followed by a discussion of antiviral activity on lines 61-71, and then returns to antibacterial activity on lines 72-74. The authors are advised to better structure the Introduction by separating these two types of activity into distinct paragraphs.

Comment 2: It is unclear why several viruses are mentioned in Section 2.5 and subsequently in Section 3, while only the HSV-1 virus remains in Sections 2.6 and 2.7.

Technical сorrections:

  1. The terms "in situ" (line 89) and "in vitro" (line 203) should be italicized.
  2. I recommend removing Figure 1, as it does not convey any substantial information. The transparency of the films can be sufficiently described in the text, which would be adequate.

Author Response

Reviewer #2

This manuscript presents a highly relevant topic. The article is well-structured and the findings are compelling, demonstrating a clear progression from concept to laboratory validation. The relevance of this research is exceptionally high. The persistent problem of healthcare-associated infections transmitted via contaminated surfaces remains a significant burden on global healthcare systems, both in terms of patient outcomes and financial costs. This issue was starkly

highlighted during the COVID-19 pandemic, which also led to a massive increase in single-use plastic waste. The novelty of the work is clearly articulated and lies in the multifunctional and sustainable approach to creating protective surfaces. The scientific significance of this study is substantial. It contributes to the growing field of advanced, functional materials by expanding the known applicability of chitosan-based materials beyond their traditional uses in packaging or wound dressings into the realm of surface disinfection and infection control. The practical implications of this research are profound and directly align with the stated goals infection control and economic benefits.

Comment 1: In the Introduction, the antibacterial and antiviral activities of the chitosan-nanoCu film should be delineated more clearly. The text on lines 43-59 first covers antibacterial activity, is followed by a discussion of antiviral activity on lines 61-71, and then returns to antibacterial activity on lines 72-74. The authors are advised to better structure the Introduction by separating these two types of activity into distinct paragraphs.

Author’s Response: According to the reviewer's suggestion, the first paragraphs of the introduction were reorganized as follows:

  1. Introduction

Viruses and bacteria can adhere to a wide range of surfaces and spread through similar mechanisms. One of the most clinically relevant Gram-positive bacteria is Staphylococcus aureus, which can rapidly form biofilms on inert surfaces, facilitating proliferation and impairing both host immune responses and the efficacy of antibiotics [1]. Other highly pathogenic microorganisms include Enterococcus spp., Escherichia coli, and Klebsiella spp., which are particularly associated with infections in urinary tract devices, prosthetic joints, and intravascular devices [2]. In a recent review article, Porter et. al systematically searched Ovid MEDLINE, CINAHL and Scopus databases for studies that described the survival time of common nosocomial pathogens in the environment. Pathogens included in the review were bacterial, viral, and fungal. They informed that common pathogens of concern to infection prevention and control, can survive or persist on inanimate surfaces for months [3].

Viral infections remain a persistent public health concern, often resulting in substantial medical and economic burdens. Furthermore, there are viral emergencies and re-emergencies for which there is currently no effective vaccines, antiviral treatments or preventive strategies, underscoring the urgent need for tools that can mitigate their spread. Although direct person-to-person transmission remains the primary route for viruses such as SARS-CoV-2, indirect transmission via contact with contaminated surfaces has also been documented [4,5]. Surfaces with a constant viral load, especially in high-traffic areas, can induce infections, where the probability can increase due to high exposure to a large number of people. Experimental studies have shown that SARS-CoV-2 can survive on various materials for extended periods: up to three days on plastic and stainless steel, 24 h on cardboard, and four hours on copper [6]. Moreover, the virus was detectable in droplets suspended in the air for up to three hours. So, the antiviral surface treatments can help to reduce or even avoid the viral transmission. This data supports the need for a risk-based approach to cleaning and disinfection practices, accompanied by appropriate training, audit and feedback which are proven to be effective when adopted in a ‘bundle’ approach [7].

Consequently, having instruments able to stop the propagation of microorganisms is a crucial issue. Currently, this poses a significant challenge because many innovations aimed at replacing everyday items are emerging, such as metal-based hospital devices [3], antimicrobial paints [3–5], and replaceable surgical instruments [6], among others.

As a result of this reorganization, the numbering of some bibliographical citations was changed.

Comment 2: It is unclear why several viruses are mentioned in Section 2.5 and subsequently in Section 3, while only the HSV-1 virus remains in Sections 2.6 and 2.7.

Author’s Response: According to the reviewer's concern, in the 3.3 section was added the next sentence:

3.3. Persistence of virucidal activity on spray-coated surfaces

One of the major challenges in the development of antiviral surface coatings is achieving long-lasting, reusable systems capable of continuously reducing the risk of infection and transmission [6]. To achieve this objective and the following ones, each surface was inoculated with the enveloped virus HSV-1, selected as a representative model due to the comparable inhibitory response observed among the enveloped viruses tested (Figure 3), whose incubation time is shorter than the other enveloped viruses, such as BCoV, ZIKV or RSV (supplementary Table 1), and because that the greatest antiviral effect of CH.CA@Cu is on enveloped viruses.

Supplementary Table 1. Characteristics of the viruses employed.

Virus

Envelope

Genome

Incubation Times (Days)

HSV-1

Enveloped

DNA

2

HSV-1 tk-

Enveloped

DNA

2

HSV-2

Enveloped

DNA

2

RSV

Enveloped

RNA

5

BCoV

Enveloped

RNA

3

ZIKV

Enveloped

RNA

5

ADV

Non-enveloped

DNA

7

PV-1

Non-enveloped

RNA

2

Technical сorrections:

  1. The terms "in situ" (line 89) and "in vitro" (line 203) should be italicized.

Author’s Response: ok, they were corrected in the text.

  1. I recommend removing Figure 1, as it does not convey any substantial information. The transparency of the films can be sufficiently described in the text, which would be adequate.

Author’s Response: Thank you for your suggestion, now the Figure 1 of the original manuscript is included as Figure 1 of the supplementary material in the revised manuscript.

Supplementary Figure 1. A) uncoated polypropylene Petri dish surface. B) film-coated polypropylene Petri dish surface.